# Perceptions of pre-exposure prophylaxis among sexually active adolescent girls and young women in Zimbabwe–A qualitative study

Kudzai Chidhanguro[1]*, Getrude Ncube[2], Owen Mugurungi[2], Wellington Murenjekwa[1], Lindiwe Mancitshana[1], Fadzai Masiyambiri[1], Sharon Munhenzva[1], Frances M. Cowan[1,3], Amon Mpofu[4], Isaac Taramusi[5], Euphemia Lindelwe Sibanda[1,3], Valentina Cambiano[6]

1 Centre for Sexual Health and HIV/AIDS Research Zimbabwe, Harare, Zimbabwe, 2 Department of AIDS and TB Unit, Ministry of Health and Child Care, Harare, Zimbabwe, 3 Department of International Public Health, Liverpool School of Tropical Medicine, Liverpool, United Kingdom, 4 National AIDS Council, Harare, Zimbabwe, 5 UNAIDS UCO, Harare, Zimbabwe, 6 Institute for Global Health, University College London, London, United Kingdom

* kudzai.chidhanguro@ceshhar.org

## Abstract

Adolescent girls and young women (AGYW; aged 15–24 years) in Zimbabwe face high risk of contracting HIV. Despite proven effectiveness of Pre-Exposure Prophylaxis (PrEP), its uptake and continuation among AGYW is low. We aimed to explore views on PrEP, norms around sexual behaviour and how programs can improve PrEP uptake among this group. From December 2021 to March 2023 nine focus group discussions were held with 92 (8–12 per group) sexually active AGYW, purposively selected from programs offering sexual and reproductive health (SRH) services to AGYW in Harare, Mazowe and Matabeleland South. Discussions were participatory and analysed thematically. Participants were aged 16–24 years; 73% had attained some high school education, 9% had never tested for HIV and 70% never used PrEP. Across all groups there was recognition that AGYW are at risk of contracting HIV. Knowledge of PrEP varied, with AGYW enrolled in the national sex worker and SRH programs more knowledgeable than the rest. A recurrent theme was that PrEP was viewed as undesirable due to its association with anti-retroviral treatment and risky sexual behaviours. AGYW thought their parents and partners would find their use of PrEP unacceptable. Unmarried sexually active AGYW seemed the most vulnerable and at high risk of HIV acquisition. Issues identified as important for improving uptake and adherence included ensuring private, confidential and free services, education of parents and partners on SRH issues and friendly attitudes of health workers. Additionally, AGYW indicated preference for the long-acting PrEP formulations, viewed as convenient and more private. Despite the recognition of being at risk of HIV, HIV related stigma and concerns about being viewed as having risky sexual behavior prevent them from taking PrEP. There is need to design youth-friendly PrEP programs

**Data availability statement:** This is a qualitative study, and we have provided de-identified focus group discussions (FGDs) transcripts (as supplementary information) that shaped the conclusion of the manuscript.

**Funding:** The study was supported by Medical Research Council [grant number MR/T042796/1] to VC. The funders had no role in the study design, data collection and analysis, decision to publish or preparation of the manuscript.

**Competing interests:** The authors have declared that no competing interests exist.

that uphold privacy & confidentiality, prevent stigma and minimize opposition from parents and partners.

## Introduction

Despite the global decline in HIV infections, inequalities exist, with some population groups such as young people showing high HIV incidence and poor engagement with health services [1]. According to The Joint United Nations Programme on HIV and AIDS (UNAIDS) 2024 report globally every week 4000 adolescent girls and young women (AGYW) aged 15–24 years acquire HIV [2,3]. In East, West, Central and Southern African nations in 2022 women and girls accounted for 62% of all new HIV infections [2,3], and AGYW are three times more likely to acquire HIV than adolescent boys and young men [1,4–9]. In Zimbabwe AGYW account for 22% of the new HIV infections and about 60 AGYW acquired HIV every week in 2024 [10].

Factors putting AGYW at higher risk of HIV are biological, socio-behavioural and structural [4,6]. Biologically, women have a higher risk of HIV acquisition per vaginal sex act compared to men, and the risk is higher at a younger age [6,11]. Engagement in intergenerational and/or transactional sexual relationships with older men, who in most cases engage in risky sexual relationships [12,13] and with whom condom use negotiation may be more challenging, exposes AGYW to high risk of getting HIV [1,6,8]. In-addition, poverty, low levels of education, gender-based violence and early marriages continue to expose AGYW to HIV. According to the Zimbabwe 2022 census, AGYW are getting married at early ages, with 1% of women getting married or in union before the age of 15 years and 18.9% before attaining the age of 18 [14]. The 2024 Zimbabwe Demographic and Health Survey found that 23% of women aged 15–19 have ever been pregnant demonstrating that a high number of AGYW faces a risk of getting HIV through having condomless sex that resulted in these pregnancies.

To reach the Global AIDS target of 95% of people at risk of getting HIV having access to HIV prevention options by 2030, further effort is required [1,15]. Pre-Exposure Prophylaxis (PrEP) for HIV is a key component of combination HIV prevention [15] and holds great potential for reducing HIV incidence amongst AGYW. Currently PrEP is available as oral formulations, dapivirine vaginal ring (DVR), and injectable cabotegravir (CAB-LA) and lenacapavir. According to the UNAIDS 2024 global AIDS update, around 3.5 million people were using PrEP in 2023 but the number is still short of the global 2025 target of 21.2 million people [16]. Zimbabwe has rolled out PrEP since 2016, and this is being offered as part of a comprehensive HIV prevention package for people at substantial risk of HIV infection, including sex workers, young women selling sex, sero-discordant couples and adolescent girls and young women [17,18]. The country's 2022 guidelines for HIV prevention outline the eligibility rules for PrEP use as follows: i) HIV negative and sexually active AND has any of the following: a) Vaginal or anal intercourse without condoms; b) A sexual partner of unknown HIV status; c) A sexual partner with one or more HIV risk factors; d) A history of sexually transmitted infection by lab testing or self-report or syndromic treatment

of sexually transmitted infection; e) Any use of post-exposure prophylaxis (PEP) [19]. In Zimbabwe, the primary donors for HIV programming including PrEP provision at the time of the focus group discussions (FGDs) were The United States President's Emergency Plan for AIDS Relief (PEPFAR) and Global Fund [20]. Oral PrEP is accessed through public and not-for-profit private sectors in Zimbabwe [20]. In most health facilities oral PrEP is accessed at no cost, however there are some council-run health facilities that requires an adult to pay a clinic consultation fees, usually $5 before accessing health services [21]. In Zimbabwe, dapivirine vaginal ring and CAB-LA were being rolled out under implementation research to inform policy and scale-up [22,23], with prioritisation of subpopulation at high risk including AGYW. CAB-LA was being implemented in 15 sites, with 6 being supported through the catalyzing access to new prevention products to stop HIV (CATALYST) and 9 through PEPFAR. However, the CATALYST trial has been stopped due to United States Agency for International Development (USAID) funding cuts [24,25] as Zimbabwe is amongst the countries that have been recently affected by the USAID funding cuts [26]. In 2025, PrEP services have been severely affected due to heavy reliance on USAID funding, with availability primarily focused on pregnant and lactating mothers [27]. For Zimbabwe this has already affected access to PrEP to non-pregnant or lactating sexually active AGYW at risk of acquiring HIV as most organisations providing HIV services have stopped working. In particular the Determined, Resilient, Empowered, AIDS-Free, Mentored and Safe (DREAMS) program has been halted [26,28]. DREAMS served some African countries representing over half of all infections occurring among AGYW globally including Zimbabwe. Its core aim was to address structural drivers to HIV including poverty, gender inequality, sexual violence, and lack of education and promoting HIV prevention (including PrEP access) amongst AGYW [29,30]. The government of Zimbabwe is making efforts to mitigate the impact of reduced international funding for HIV programs including PrEP [31,32]. World Health Organisation has approved Zimbabwe, along with other African countries including South Africa, Zambia, Uganda, Tanzania, Nigeria, Kenya, Botswana to begin receiving Lenacapavir by 2026 [32,33]. Before the recent funding cuts, low PrEP uptake was being witnessed amongst AGYW regionally and locally in Zimbabwe [34], with only 6% of the surveyed sexually active AGYW at risk of HIV found to be on PrEP in some provinces [35]. This is largely due to challenges with oral PrEP adherence, unfriendly youth PrEP services and HIV related stigma [36–39]. There is an urgent need to scale up well-designed and effective programs that meet the HIV prevention needs of AGYW at substantial risk of HIV acquisition and enhance uptake and adherence to PrEP.

The aim of the study was to inform the development of an acceptable, context-relevant program for facilitating uptake of PrEP for AGYW. To achieve the aim of the study we conducted a qualitative study exploring perceptions of sexual HIV risk among AGYW in Zimbabwe and generating broad overviews of PrEP acceptance and barriers.

## Methods

### Ethical statement

The study was approved by the Medical Research Council of Zimbabwe (MRCZ) (REF: MRCZ/A/2760), the Research Council of Zimbabwe, and University College London Research Ethics Committee (REF: 21017/001). Waiver of parental consent was obtained from all ethical committees for participants aged 15–17 years old to maintain confidentiality about being sexually active and minimise potential social harms. Written informed consent was obtained from all participants before study procedures were conducted.

### Inclusivity in global research

Additional information regarding the ethical, cultural, and scientific considerations specific to inclusivity in global research is included in the Supporting Information (S1 Checklist).

### Study population, location and sampling

We conducted nine FGDs with sexually active AGYW in Zimbabwe. This was part of a mixed-methods study [35,40] aimed at informing the development of an intervention for improving PrEP uptake amongst sexually active AGYW. We

defined the criteria for participation in the FGDs as sexually active AGYW, defined as having had sex with a man in the past 12 months (with or without a condom), aged between 15–24 years. FGDs were held in urban and rural settings of three provinces, namely Harare, Mashonaland Central and Matabeleland South. All FGDs were held in-person with both the participants and moderator being physically present in each of the province stated above during the discussions. Purposive sampling was done to ensure representation of sexually active AGYW within the ages of 15 and 24 who are enrolled at universities, those who identify as female sex workers, pregnant and lactating AGYW accessing antenatal care and youth centres services. Youth centres serve vulnerable young people, providing them with vocational training skills, education on integrated HIV prevention, sexual and reproductive health rights and sexual and gender-based violence services [41,42]. According to literature these groups of AGYW are perceived to be at a high risk of getting HIV. Moreover, sexually active AGYW receiving sexual and reproductive health and HIV prevention services were also included. We engaged these programmes/organisations/clinics in the recruitment of the FGD participants as they were already working with sexually active AGYW offering them different sexual and reproductive health services. This was done to maintain the confidentiality of their sexual behaviours.

### Data collection

From 10 December 2021 to 13 April 2022 seven FGDs were conducted in Harare and Mashonaland Central provinces and two more FGDs were done from 22 to 23 March 2023 in Matabeleland South. During data analysis discussions it emerged that it would be helpful to get insights from another province (Matabeleland South) where there is higher vulnerability to HIV among AGYW [43].The over one-year gap before the focus group discussions in Matabeleland South was due to unforeseen delays in obtaining required community entry approvals. FGDs were split by age group, 15–19 years old, and 20–24 years old to ensure participants were able to discuss general sexual behaviour issues with peers of the same age [29,30]. Each FGD was conducted with a group of AGYW with similar demographics, for example young women selling sex participated in their own FGD separate from the young AGYW recruited from the antenatal clinics. Program/ institution staff referred potentially eligible and willing participants to study staff.

Discussions were held in places that offered privacy and were easily accessible to participants. The FGDs were moderated by trained female social scientists with experience of conducting FGDs and working with young people. A participatory approach was used where participants firstly engaged in setting up the ground rules where issues of confidentiality, respecting each other's views and speaking through the moderator were raised and agreed upon. As a common practice in participatory approaches, PrEP information was provided to the participants in the first instance before the actual discussion commenced [44]. This ensured that participants have a shared understanding of PrEP which led to a more focused, richer discussions. The discussions were also open, taking a less structured approach with the moderators ensuring a free-flowing natural, but focused communication. A participatory approach of having interactive group activities was achieved using role plays that stimulated and encouraged discussions amongst the participants.

Three role plays were conducted in each FGD, creating 3 groups for each role play in each FGD. In each FGD, two to three participants acted a scene depicting conversations AGYW would have with friends or male partners about taking oral PrEP, which was followed by detailed discussion to explore views/understanding of the group [45,46]. Discussions in these small groups for role plays lead to some shy participants opening up and makings contributions. S1 Appendix presents the discussion guide with details of scenarios that were acted out. The questions centred around perception of HIV risk, PrEP knowledge and understanding, identifying barriers and facilitators to PrEP uptake and the preferred PrEP service delivery models amongst sexually active AGYW. Detailed discussions were made after each role play to elicit views, identify different views within the group [45,46]. Participatory approaches enable participants to open up, relax, interact and contribute comfortably and can avoid domination of discussions by a few people [45,46]. During the discussions participants were shown oral PrEP, vaginal ring and pictures of CAB-LA and given information on how they work and their different efficacy levels. FGDs were moderated by trained qualitative researchers in the local language, either Shona

(KC) or Ndebele (LM). Discussions lasted between 60–90 minutes and were audio-recorded. All discussions were held in the context of the COVID-19 pandemic, and we employed relevant procedures to prevent COVID-19. Each participant received $5 as reimbursement for the time they took participating in the FGDs.

### Data handling and analysis

All FGD audio records were transcribed verbatim and translated to English by four qualitative researchers (KC, LM, FM & SM), then deleted from the recorders. All audio-records and transcripts were de-identified by assigning unique study participant numbers. Data analysis followed the principles of Thematic Analysis [47]. Soon after each FGD, detailed field notes were written with attention to emerging themes. We held regular discussions among the study team to familiarise with the data, interrogate findings and inform further exploration. Team discussions drew comparisons within and across participant groups. It was during these discussions that it emerged that it would be helpful to get insights from another province (Matabeleland South) where there is higher vulnerability to HIV among AGYW [43]. The team discussions and the field notes were used to code the data; identifying specific pieces of content relevant to the research topic and later used to develop and refine a coding framework [47,48] which was used for coding the transcripts in NVIVO 14. The identified codes were organised into themes connecting related concepts. Themes were then reviewed; understanding the meaning of each theme and finally refined.

### Theoretical framework

There is growing recognition in the literature that, besides individual behaviour, broader environments in which people live and work in largely contributes to their decision-making in utilising health services. The social ecological model (SEM) illustrates interactions between characteristics of the individual, community and the environment [49] that influences health behaviour. This framework recognizes that health experiences and outcomes are often influenced by characteristics situated within and beyond the individual [50,51]. The SEM levels used are described as; (i) individual (intrapersonal) level such as knowledge, attitude and beliefs; (ii) interpersonal level describing the influence of social interactions on health decisions. This may include the immediate physical environment and social networks in which an individual live including family, friends and peers; (iii) organisational level such as formal, informal organisations and health provider aspects and (iv) community level where communities may share cultural, ethnic and religious characteristics [39,52]. We used this construct to understand barriers and facilitators to PrEP uptake among sexually active AGYW. The SEM framework served as a guide addressing how different elements influence PrEP uptake across multiple levels and was incorporated in our analysis. In refining the identified themes, we examined how the themes were linked to the theories and constructs of the SEM framework.

## Results

Overall, 92 (8–12 per group) participants took part in the FGDs. Participants were aged 16–24 years; 70% (64/92) had attained some high school education, 8% (7/92) had never tested for HIV and 50% (46/92) had never used PrEP (Table 1). We recruited AGYW who were enrolled or engaging with the National Key Populations program (2 FGDs - 16 AGYW), the DREAMS program (2 FGDs - 21 AGYW), universities (1 FGD - 9 AGYW), youth centres (1 FGD - 12 AGYW) and ante-natal care clinics (3 FGDs - 34 AGYW).

We present the main results on the perceptions of PrEP among AGYW, highlighting important elements that influences PrEP uptake amongst AGYW. Guided by the Social Ecological Model framework, we present these elements as intrapersonal (individual), interpersonal, organizational and community. Fig 1 shows how the various levels of influence, from individual to community, interact to affect an individual's decision to use PrEP.

### Intrapersonal (individual) level

Issues that served as barriers and facilitators to PrEP uptake at intrapersonal level included: HIV risk perception, PrEP knowledge and oral PrEP packaging.

**Table 1. Characteristics of study participants (n = 92).**

| Characteristics | Category | Number of participants |
|---|---|---|
| Age range (years) | 16-19 | 42 (46%) |
| | 20-24 | 50 (54%) |
| Current marital status | Married | 31 (34%) |
| | Single and never married | 58 (63%) |
| | Single: Separated or Divorced | 3 (3%) |
| Number of children | None | 39 (42%) |
| | Currently pregnant | 5 (6%) |
| | One | 36 (39%) |
| | Two or more | 12 (13%) |
| Occupation | Unemployed | 60 (65%) |
| | Student | 10 (11%) |
| | Formal Employment | 8 (9%) |
| | Self-employment | 14 (15%) |
| Highest level of education completed | No education | 2(0.02%) |
| | Primary | 14 (15%) |
| | Secondary incomplete | 20 (22%) |
| | Secondary | 44 (48%) |
| | Tertiary | 12 (13%) |
| History of HIV testing | Never tested | 7 (8%) |
| | 1-5 times | 66 (72%) |
| | 6-10 times | 13 (14%) |
| | 11-20 times | 6 (6%) |
| History of PrEP use | Have never used PrEP before | 46 (50%) |
| | Have used PrEP before but not currently | 19 (21%) |
| | Currently using PrEP | 1 (1%) |
| | Not asked | 26 (28%) |

## Awareness of high HIV risk amongst adolescent girls and young women

Across all groups there was recognition that AGYW are at a high risk of contracting HIV; with description of three issues that expose them to this risk. Firstly, AGYW were reported to engage in condom-less transactional sexual relationships with much older men, commonly known as "blessers", who they suspect are either living with HIV or at high risk of getting HIV. Relationships with blessers were believed to be mostly associated with sexually active AGYW in colleges and those who identify as female sex workers. The young women say they do not have power to negotiate for condom use in this situation because of the age difference and fear of loss of financial benefits associated with the sexual relationship. There was also belief that having condomless sex is a sign of love.

> "There is no way you are going to have sex with him using a condom because at the end of the day he is saying I am giving you raw money so you must give me raw sex." **(21 years old university student)**

> "So, (he would say) *I want to sleep with you without protection for you to see that I love you. Plus, what is it that you do not have, I am giving you all the money.*" **(17 years old DREAMS beneficiary)**

Secondly, the groups identified multiple and concurrent sexual partnerships by male partners of AGYW as another risk factor for getting HIV. It was widely believed that AGYW were generally not aware of their partners' HIV

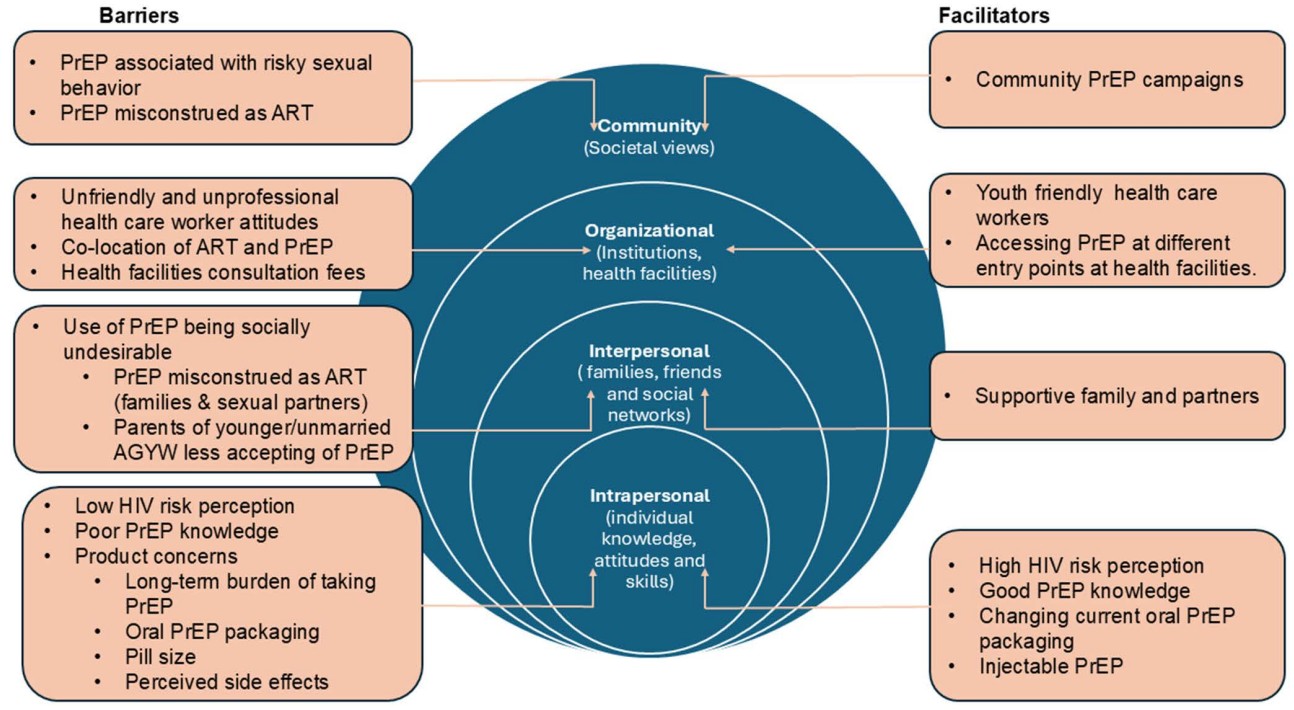

**Fig 1. Factors influencing PrEP uptake among sexually active AGYW at each level of the socio-ecological model.**

status. Additionally, there was belief that many AGYW were in relationships with men who had multiple concurrent partners.

*"Just imagine if he has five people that he blesses. If he has sex with all these five without protection, all of you will contract it."* **(20 years old female sex worker)**

Thirdly, sexual abuse by relatives or strangers was highlighted to be putting AGYW at risk of getting HIV as indicated in this scenario:

*"Maybe your father is late, and you are staying with your mother's brother. He is a breadwinner at the home, then he would take you and say for him to do everything here, taking you to school, buy you food, you must have sex with him."* **(19 years old youth centre client)**

However, there was evidence that some AGYW had low perception of HIV risk. Although they acknowledged that AGYW engage in relationships with older men for financial benefits, they did not seem able to articulate/appreciate how that increased the risk of getting HIV. This lack of understanding was most pronounced among the pregnant and breastfeeding AGYW who were recruited through the antenatal or postnatal clinics in the rural areas. High risk perception was thought to influence one's decision making in using PrEP as they acknowledged that they were at a risk of getting HIV.

### Varied knowledge of PrEP

Knowledge of oral PrEP varied amongst the participants: from very good understanding to partial to no knowledge at all. For vaginal ring and CAB-LA most of them were hearing about them for the first time and the few participants who had heard about

them lacked in-depth details of the efficacy levels and how they are used. Participants enrolled in the National Key Populations program, the DREAMS programs, and those at universities had better knowledge than most of the participants from antenatal clinics who had poor or no knowledge at all, with many of them hearing of PrEP for the first time. The National Key Populations and DREAMS program prove comprehensive PrEP education [53] as do organisations like Students And Youth Working on reproductive Health Action Team who offer sexual and reproductive health education in tertiary institutions [54]. For those who did know about PrEP, they viewed it as a good thing for AGYW. Good knowledge of PrEP was perceived to facilitate PrEP uptake as AGYW indicated that understanding PrEP's effectiveness in HIV prevention increased one's confidence in using it.

*"I think the number 1 is the protection from HIV and all of us here we are afraid of it so that's the main thing." **(22 years old university student)***

We found that one group of AGYW remain particularly vulnerable, unsupported and at a great risk of poor engagement with HIV services; the unmarried, sexually active AGYW who were not currently engaged in HIV/sexual and reproductive health programs. FGDs showed they had sexual and reproductive health and HIV knowledge gaps and no supportive structures for discussing their sexual and reproductive health needs because of the deeply enshrined societal stigma associated with sex outside marriage [55]. In contrast, married AGYW not engaged in programs reported being able to get support on sexual and reproductive health issues from their mothers/family.

### Product concerns

**Long-term burden of taking PrEP.** Some shared that the long-term use of PrEP can be perceived to have the same burden as taking HIV treatment known as antiretroviral therapy (ART), describing that it seemed pointless to take lifelong PrEP in order to avoid taking lifelong ART. Others indicated that the idea of taking pills daily maybe cumbersome making it difficult to adhere consistently to taking PrEP.

*"Taking pills is boring so they would be feeling like they now seem like the people who are positive because the ones who are positive take them every day so it's now just the same…." **(21 years old university student)***

**Oral PrEP pill size and packaging.** In addition, some thought the oral PrEP tablets were too big and difficult to swallow. There were also concerns of the rattling sound of the oral PrEP packaging that draws unneeded attention making it difficult for one to take the pills discreetly. Most of the AGYW shared that changing the current oral PrEP packaging could be helpful and making it more appealing and improving PrEP use among AGYW. Some suggested using the plastic pill bags that are commonly used for dispensing pills in the country.

*"It's being said aah PrEP, the pill is too big, it's difficult to swallow, maybe that is why she was refusing…because it's as big as the HIV one so it's difficult to swallow." **(17 years old female sex worker)***

*"If they change those containers, even if they put in plastic packaging maybe it can work." **(17 years old female sex worker)***

**Perceived side effects.** Based on personal experiences and hearsay, the groups mentioned side effects which included nausea, tiredness, loss of appetite and weight gain. It was thought that fear of possible side effects could deter potential users from starting PrEP.

*"Others who would have taken it, may be telling you the symptoms that if you take it, you will feel a loss of strength. You… you start feeling weird, others start to gain weight so one will now be thinking that aah I will start looking like a sick person." **(20 years old female sex worker)***

 

## Preference for Injectable cabotegravir (CAB-LA) & Dapivirine vaginal ring

Participants seemed to be quite positive about the CAB-LA and vaginal ring. These were highly preferred and were cited as potential facilitators to improve PrEP uptake as they offer privacy and convenience, as one does not need to take them daily and has a longer dispensing frequency.

*"If I have my ring, I don't have to worry that mum will come across the pills."* **(24 years old female sex worker)**

Both CAB-LA and vaginal ring were thought to provide a discreet option particularly to AGYW under parental care or living with partners as some shared that some partners might take the tablets away.

*"Yea those types are good because let's say it's a woman who is in a situation where her pills (oral PrEP) are burned or thrown away. If I get the injection, I will be safe, no one affects it."* **(19 years old DREAMS beneficiary)**

Looking at vaginal ring versus injectable, it was discussed that even if both offer privacy and confidentiality of using PrEP, the injectable seemed to be preferred as it is more discreet unlike the ring, which they worried someone can feel during sexual intercourse. Most seemed to be concerned with possible side effects of using the ring rather than the injectable, with evidence of myths that surround products that are inserted into the vagina.

*"I may go and get the ring and will be getting it every month; doesn't it have any side effects that all of a sudden, I will be told you have cancer, you have uterine cancer."* **(19 years old youth centre client)**

## Interpersonal level

At interpersonal level lack of support from family and sexual partners were perceived to be barriers to PrEP uptake amongst sexually active AGYW.

### Use of PrEP being socially undesirable

A recurrent theme among participants was that PrEP was viewed as undesirable due to its association with being sexually active, having risky sex and being on anti-retroviral treatment. For this reason, AGYW thought their parents, sexual partners and peers would find their use of PrEP unacceptable.

**Fear that PrEP can be misconstrued as HIV treatment.** Most of participants raised that oral PrEP and ART look similar, resulting in PrEP being misconstrued as HIV treatment. They said that PrEP and ART are often collected from the same access points at the health facilities, and so this reinforces the assumption that one is living with HIV and taking ART. Access of PrEP at different entry points at health facilities was viewed to be a potential facilitator to PrEP uptake.

*"I will be seen holding my bottle, people do not know that the bottle with a red line…, is it maroon, its PrEP. The one with a blue line is for HIV…Someone who doesn't know, if they see me holding that container, she will spread that saying have you seen her, she is taking these tablets."* **(19 years old youth centre client)**

*"If they keep on giving us the pills in the same room with people who have HIV, people will not agree. They will think that we are on ART treatment."* **(17 years old DREAMS beneficiary)**

**Use of PrEP could negatively affect relationship with sexual partner.** A major concern among participants was the perception that using PrEP could jeopardize relationships with their sexual partners. It was feared that if a male partner learned that their female partner was on PrEP, he would assume she has other sexual partners. Married women who were

worried about their husband's risk taking felt they would need his approval to take PrEP, although they felt it was unlikely to be given for the reasons above.

*"Some men might turn it against you that you are being promiscuous, why are you taking the pills." **(21 years old university student)***

**Parental influence.** Most of the participants reflected on how parents are influential in one's decision to take PrEP. They shared that in most instances parents of unmarried AGYW find sex before marriage unacceptable, hence taking PrEP (which is thought to be for "promiscuous" people), would be extremely unacceptable. Furthermore, they highlighted fear of negative consequences if the parent discovered use of PrEP, such as being chased away from home (to go and live with the sexual partner). Generally, AGYW felt their parents/guardians did not have positive attitudes to AGYW protecting their sexual and reproductive health, hence fail to offer the necessary support to them.

*"When we look at the girls who are still living with their mothers for you to be seen holding the bottles of pills…To tell them that it's PrEP that is used for this…, they will say are you going to the beerhall* [figurative term for engaging in sex work]*, so it discourages them…" **(21 years old female sex worker**)*

*"Our parents believe that we are 100% pure, that we are not even sexually active. So, if you are seen with PrEP or condoms, you will pack your bags and be left with a guy you don't even love." **(21 years old university student**)*

## Organizational level

At organizational level concerns related to health facilities and health facilities consultation fees hindered PrEP uptake amongst sexually active AGYW.

### Unfriendly and unprofessional health care worker attitudes (in some public health facilities)

Judgmental, unfriendly and unprofessional attitudes by some health care workers towards AGYW accessing PrEP and other sexual and reproductive health services were reported to be a barrier to PrEP uptake. This was mostly reported by AGYW based on their peers or their experiences with health care workers when accessing PrEP or sexual reproductive health services at public health facilities. Some health care workers were thought to be unaccepting of sexually active adolescents, with reports that they would embarrass them by questioning them on why they were sexually active particularly if they were unmarried. Participants reported that these confrontations could be in public, disclosing to everyone present that the young person is sexually active. With this regard, AGYW highlighted the importance of having friendly health care workers with positive attitudes towards AGYW accessing HIV prevention and sexual and reproductive health services.

*"They come here, and they are unfriendly towards us, you even think it is better if I stop using this PrEP whatever happens, happens! So, we want someone who will be smiling and happy." **(19 years old DREAMS beneficiary)***

*"Even in the hospitals, let us say I am 16 years or 15, but I know fully that I am already engaging in sexual activities. I come to the hospital and say I want PrEP; they will start saying at your age why do want PrEP." **(19 years old DREAMS beneficiary)***

Participants explained that similar problems are encountered when AGYW need to access treatment for sexually transmitted infections:

*"Such a young child like you, you actually want us to take your panties off and examine you, such a young child like you doing adult-related things." (21 years old university student)*

*"I have an STI, and I want to take PrEP, maybe I am 15 years or 16 years, they may call each other, the nurses three or four…Come and see what is here…Come and see this young child…Come and see the genitals of this child…They would start laughing, sometimes they would start scolding you." (19 years old youth centre client)*

In addition to the problem with co-location of ART and PrEP clinic mentioned above, there was also worry that health care workers would divulge one's decision to take PrEP to their parents. This was a challenge where the young person and their family were known to clinic staff.

*"Aah if they are local clinics and you stay there, if you get PrEP, before you get home your mother would have been told about it by the nurses from there." (21 years old female sex worker)*

### Health facilities consultation fees

Currently oral PrEP is being offered for free in Zimbabwe, including in donor funded and run institutions, however at some public sector health facilities one needs to pay a consultation fee of $5 to get the PrEP prescription. Participants said that this is not affordable for most AGYW, especially those that are still under parental/guardian care.

*"…you cannot enter the clinic and get the service even for free without paying…, that fee.... consultation. Even if you know I want to go and get PrEP that is offered for free. They will tell you first pay that $5 then come and collect..." (21 years old university student)*

### Community level

At community level the association between PrEP and risky sexual behaviour and PrEP being misconstrued as ART mentioned at the interpersonal level were perceived to be barriers to PrEP uptake amongst sexually active AGYW.

### PrEP associated with risky sexual behavior

Participants reported that communities assume that anyone who uses PrEP engages in risky sexual behaviors, that is that they have more than one sexual partner or engage in condomless sex, often labelling PrEP users as "loose" (promiscuous). There were reports that some communities thought PrEP was offered only to sex workers.

Participants suggested having community PrEP campaigns, disseminating correct and accurate PrEP information and combating community PrEP misconceptions.

*"You just come even at XXX, moving around with your car. Educating people about*

*PrEP…." (19 years old youth centre client)*

### Discussion

In this qualitative study among sexually active AGYW, we explored perceptions on PrEP. Various determinants at the four levels of the Social Ecological Model framework were found to influence PrEP uptake amongst this population. We found that most had good knowledge of HIV risk, while PrEP knowledge varied, with the majority having good understanding of PrEP related issues. Unmarried AGYW who were not attending HIV/SRH programs had the least knowledge of PrEP

and HIV risk. Reported barriers to PrEP use included moral judgement, fear of being judged to be 'promiscuous', barriers related to the product, e.g., pill burden, packaging – which was thought to be similar to that for ART, and fear of possible side effects. Health facility barriers included unfriendly/unprofessional health care worker and user fees. Participants reported strong preference for long-acting PrEP products, particularly the injectable, because they offered more privacy and convenience.

This study contributes to the growing literature on the barriers to PrEP uptake amongst AGYW in Southern and East Africa [38,56–59] which includes PrEP being socially undesirable, PrEP product and health facility concerns. Particularly from Zimbabwe though there are limited studies reporting on barriers to PrEP uptake amongst sexually active AGYW, two studies have explored views on barriers to PrEP use amongst AGYW, [36,59] similar insights on the oral PrEP package and unfriendly health workers were reported. This study provides new insights on the vulnerability of unmarried sexually active AGYW and exposure to high risk of getting HIV.

At intrapersonal (individual) level issues that have been reported by AGYW of oral PrEP pills being too big, uncomfortable to swallow, possible side effects and the rattling sound from the bottles are consistent with previous literature [36,59,60]. In Zimbabwe most medication is packaged in small medicine boxes or plastic zip lock bags and only a few, including ART, are packaged in bottles, raising curiosity around having a bottle with medication. Skovdal 2022 acknowledges efforts being made by providers to change PrEP packaging as is being done through the 'V' initiative, where PrEP is repacked in containers that look like lip-gloss [36,61,62] http://www.conrad.org/launchingv. The 'V' initiative is a brand and service delivery strategy that was designed to increase oral PrEP uptake and continuation among AGYW [61]. From the early implementation learnings from Zimbabwe [61], they found that the V's lip gloss shaped pill case provided discretion and made it easier to carry increasing PrEP uptake amongst AGYW. Changing the PrEP packaging also addressed issues related with PrEP being socially undesirable when misconstrued as ART [36]. It is important to differentiate packaging of PrEP from ART and make oral PrEP packaging to being more discreet, fun and engaging to increase its uptake amongst the AGYW, to preventing stigmatization [63].

The introduction of CAB-LA has potential to cause a step change in PrEP uptake as it addresses many of the barriers associated with oral PrEP, including daily pill taking. CAB-LA has been reported to be highly acceptable as it offers convenience and privacy [64–66]. It has been recently introduced in Zimbabwe and other African countries through implementation studies mostly for female sex workers [25,67], however its implementation has been interrupted and slowed down since the USAID funding cuts in 2025. Across the groups, AGYW indicated that they would prefer CAB-LA as it offers convenience, privacy, discretion and mostly importantly no daily adherence. To note neither of these technologies (CAB-LA and vaginal ring) were available to AGYW at scale at the time of the discussions. Another game changer for PrEP is the newly introduced 6 monthly injectable Lenacapavir [68]. Lenacapavir will likely be even more preferred and have better adherence, given its long duration of action [69].

At interpersonal level, unmarried sexually active AGYW seemed most vulnerable and at high risk of getting HIV as they often lack support from parents, health care workers and the community. Skovdal et al [58,70] shared similar insights with parents in Zimbabwe having strong cultural and religious beliefs making it taboo for a girl to be sexually active before marriage. Research from Tanzania [38] also found lack of support for the unmarried AGYW, their married peers are accorded some respect based on one's marital status and have a platform to openly discuss and be involved in sexual and reproductive health issues with their parents and within their communities. These findings of unmarried AGYW being at a risk of getting HIV are reflected in the estimates produced by The Incidence Patterns Modelling (National AIDS Council and UNAIDS, 2017 and 2021), which found that never married females (who are mainly adolescents girls and young women) contributed the highest proportion (24% in 2017 and 28% in 2021) of new infections [71,72].

Studies conducted in Africa exploring barriers to parent-child communication found that young people face difficulties in having dialogues about sexual and reproductive health issues [73–78]. Parents are very influential and unmarried sexually active AGYW are often still under parental care and may require their approval to access and use sexual and reproductive

health services if aged less than 16 unless they have children or are married [39,79]. There seems to be little progress on implementing programs to improve parental support at scale despite recognition of the importance of this in the Zimbabwe National Adolescent and Youth Sexual and Reproductive Health Strategy II: 2016–2020 [80]. The strategy further states that a Parent-Child Communication Package, which unpacks the roles and responsibilities of parents, guardians and communities in promoting the health care workers rights and needs of young people will be developed. There is need to co-develop interventions targeting AGYW together with the parents to support parents of AGYW.

At organisational level, unfriendly services coupled with judgemental attitudes from health care workers towards AGYW is a concern that continues to affect young women's access to SRH services, including PrEP (and will also potentially be an issue for CAB-LA). Some health care workers from Zimbabwe and Tanzania [36,38] shared the sentiments that some of their colleagues are judgemental and have unfriendly attitudes towards AGYW, emphasising the need to continue working towards provision of youth friendly services and ensuring health care workers are more accepting of sexually active AGYW whether married or not [81]. A similar observation was found in a systematic review of aspects influencing access to and utilisation of youth-friendly sexual and reproductive health services in East, West, Central and Southern African nations [82]. The issue of PrEP and ART being provided at same access point within public health facilities is similar to what Ddaaki et al. found, as they shared that accessing PrEP and ART at the same points acts as a source of stigma for potential PrEP users [83]. However, Zimbabwe Ministry of Health and Child Care made efforts to decentralise PrEP access from the opportunistic infections department, where ART is collected from. More entry points including family planning, ante-natal and post-natal care departments were identified [84]. Reflecting, our results align with findings of other studies, indicating a strong need to increase the efforts being done to resolve these barriers.

A scoping review by Ekwunife 2022, offers valuable insights that can guide strategies to promote PrEP acceptability [85]. It included interventions designed to create demand, facilitate initiation and continued use of PrEP using PrEP uptake strategies like social marketing. Such strategies can be piloted in Zimbabwe as tapping into social media, a mode of communication that appeals to young people and has potential to increase interest in PrEP.

The strength of this study is that we used robust participatory qualitative methods to explore AGYW sexual behavior, their risk perceptions and views on barriers and facilitators to PrEP uptake. We obtained data from a diverse participant group of sexually active AGYW including sex workers, college/university students, and young mothers, some married and some not. Limitations include potential for social desirability bias; however, the researchers were trained to encourage honest opinions and established rapport at the beginning and used participatory approaches that ensured that the participants were comfortable. Another limitation is that a minority of participants had not heard of PrEP, hence their views may have been more hypothetical. However, these were balanced with the views of those who had experience being offered and using PrEP.

## Conclusion

We found that AGYW face barriers to PrEP uptake despite acknowledgement of their risk of getting HIV. These include PrEP being socially undesirable, PrEP being misconstrued as ART, pill burden and unfriendly and judgmental health care worker attitudes. Importantly, unmarried sexually active AGYW who are not enrolled in donor funded programs are particularly vulnerable – they have not been reached with information and do not feel able to speak to parents/community to get HIV/sexual and reproductive health information. Addressing these barriers requires involvement of parents, male partners and health services. Future research is needed on how PrEP interventions can be tailor-made to suit AGYW at high risk of HIV; identifying delivery approaches that resonate with young people while addressing barriers to access.

## Supporting information

**S1 Appendix. Discussion guide with acted scenarios for focus group discussions.**
(DOCX)

**S1 Checklist. Inclusivity in global research.**
(DOCX)

**S1 File. AGYW-FGDs transcripts.**
(ZIP)

## Acknowledgments

We would like to thank all the study participants for their time and valuable insight.

## Author contributions

**Conceptualization:** Kudzai Chidhanguro, Getrude Ncube, Owen Mugurungi, Wellington Murenjekwa, Frances M Cowan, Amon Mpofu, Isaac Taramusi, Euphemia Lindelwe Sibanda, Valentina Cambiano.

**Data curation:** Kudzai Chidhanguro, Euphemia Lindelwe Sibanda, Valentina Cambiano.

**Formal analysis:** Kudzai Chidhanguro, Lindiwe Mancitshana, Fadzai Masiyambiri, Sharon Munhenzva, Euphemia Lindelwe Sibanda, Valentina Cambiano.

**Funding acquisition:** Valentina Cambiano.

**Investigation:** Kudzai Chidhanguro, Lindiwe Mancitshana, Fadzai Masiyambiri, Sharon Munhenzva.

**Project administration:** Kudzai Chidhanguro, Euphemia Lindelwe Sibanda, Valentina Cambiano.

**Supervision:** Kudzai Chidhanguro, Euphemia Lindelwe Sibanda, Valentina Cambiano.

**Writing – original draft:** Kudzai Chidhanguro, Euphemia Lindelwe Sibanda, Valentina Cambiano.

**Writing – review & editing:** Kudzai Chidhanguro, Getrude Ncube, Owen Mugurungi, Wellington Murenjekwa, Lindiwe Mancitshana, Fadzai Masiyambiri, Sharon Munhenzva, Frances M Cowan, Amon Mpofu, Isaac Taramusi, Euphemia Lindelwe Sibanda, Valentina Cambiano.

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
