## [Decision Letter · Decision Letter 0]

27 May 2025

PGPH-D-25-00857

Perceptions of pre-exposure prophylaxis among sexually active adolescent girls and young women in Zimbabwe – a qualitative study.

Dear Dr. Chidhanguro,

Thank you for submitting your manuscript to PLOS Global Public Health. After careful consideration, we feel that it has merit but does not fully meet PLOS Global Public Health’s publication criteria as it currently stands. Therefore, we invite you to submit a revised version of the manuscript that addresses the points raised during the review process.

We look forward to receiving your revised manuscript.

Kind regards,

Guillaume Fontaine, PhD, RN

Academic Editor

Journal Requirements:

Additional Editor Comments (if provided):

Dear authors,

Thank you for submitting this important work to PLOS Global Public Health.

The reviewers and I appreciate your manuscript. However, there are a number of aspects that will need to be addressed before moving forward with publication. First, consider selecting and clearly articulating a qualitative or implementation-science framework such as the Social Ecological Model or the Theoretical Domains Framework to organise and interpret your themes. Second, it would also be critical to expand the Methods section, including specifying age ranges and HIV-risk criteria, explaining recruitment through DREAMS and other programmes, justifying the one-year gap before the third province’s focus groups, describing reimbursement and moderator details, and outlining exactly how coding led to your final themes. Third, it would be important to provide essential context on Zimbabwe’s PrEP landscape, including funding sources, eligibility rules, consultation costs, and how the 2024–25 budget crisis may affect access; this material belongs in the Introduction, not buried in the Results. Fourth, revise the participant table to include risk factors, partner status, and prior or current PrEP use, and present any motivations or facilitators alongside the barriers you already report. Finally, please streamline the writing: spell out most acronyms, correct spacing and phrasing issues, reference policy or efficacy claims, and ensure the Data Availability statement does not jeopardise participant confidentiality.

Guillaume Fontaine

Reviewers' comments:

Reviewer's Responses to Questions

**Comments to the Author**

1. Does this manuscript meet PLOS Global Public Health’s publication criteria?

Reviewer #1: Yes

Reviewer #2: Partly

2. Has the statistical analysis been performed appropriately and rigorously?

Reviewer #1: N/A

Reviewer #2: N/A

3. Have the authors made all data underlying the findings in their manuscript fully available (please refer to the Data Availability Statement at the start of the manuscript PDF file)?

Reviewer #1: Yes

Reviewer #2: Yes

4. Is the manuscript presented in an intelligible fashion and written in standard English?

Reviewer #1: Yes

Reviewer #2: No

Reviewer #1: Thank you for the opportunity to review this manuscript reporting on AGYW's perceptions of accessing PrEP in Zimbabwe. The use of scenarios to elicit discussion worked well for this study.

I have two broad comments for the authors to consider -

1. The paper would be strengthened by use of theoretical framework or lens with which the data can be interpreted.

2. Given the drastic and appalling changes to global funding this year, is this currently happening (see lines 315-316)? It would be useful for the authors to add some contextualisation re current funding environment within the manuscript.

Also, please read through manuscript and correct spacing issues - sometimes two spaces between words, a space between a word and punctuation, or no spacing between word and reference.

Other comments/suggestions which I believe, once addressed, will strengthen the quality of the manuscript.

Define ages of AGYW in first sentence of both Abstract and Introduction.

Lines 26-27: What were the risk factors among participants - e.g., sex and/or injecting drug use, other? Were all participants sexually active or at risk of HIV exposure?

Line 29: How many participants enrolled in either of these?

Lines 30-31: It would be helpful to know their risks factors

Lines 87-89: Specify "in Zimbabwe" in this sentence

Lines 106-107: These are 4 very diverse groups - were FGDs held with participants from specific demographics or were AGYW from across these 4 groups included together in FGDs?

Lines 110-112: These needs explanation. Why was there nearly a year gap between completion of FGDs in the two provinces and conducting two more FGDs in the third province? Was the third province added later? If so, why?

Lines 112-120: This should be moved to sub-section above - Study Population, Location and Sampling

Lines 135-136: Who are Shona and Ndebele? It appears they are not included in the author list.

Line 148 "four qualitative researchers" - Are they co-authors? If yes, include their initials.

Table, History of PrEP Use "Not asked" - Is this not asked or didn't answer? If not asked, does that mean question was not asked in 2 or more FGDs. Was this intentional? I acknowledge that sometimes an interviewer can forget to ask but curious if there was a considered reason for not asking

Lines 157-159: What proportion of participants indicated they were involved in this?

Add to participant demographics how many participants had previously used (or were currently using) PrEP.

Line 193 "prove": This kind of statement should be referenced.

Line 203: Please provide a reference for this statement.

Line 378: This is a long list of refs that appear to be superficially included without much substance as to why they are referenced.

Lines 423-424: This is personal preference, but I believe this sentence could be strengthened through modifying language - perhaps, "Our results align with findings of other studies, indicating a strong need to increase the efforts being done to resolve these barriers"

Reviewer #2: Thank you for the opportunity to review this article. Please see below some general and specific comments that I hope will be of assistance to the authors to help increase clarity for international readers.

Entire manuscript: Would remove "factors identified" or use of "factors" which is more often identified with quantitative research/epidemiology analysis

Introduction: Might want to rephrase slightly: “older men, who are most likely having risky sexual relationships…” I do not think that an international audience would assume this per se, and so I would recommend that the authors rephrase.

Introduction: “Inadequate schooling” – please elaborate. Does this mean poor sexual health in school? Or fewer educational opportunities. Please rephrase.

Please provide some (brief; 1-2 sentences) of how PrEP is provided in Zimbabwe – is this through the Global Fund? National Ministry of Health?

Introduction: Is there any understanding/evidence of why the uptake of PrEP is low? Do people have to co-pay? Is it only available to those considered high-risk for HIV? Has any other research been conducted in Zimbabwe in this area?

Introduction: I would suggest revising the last sentence to state the aim of the study. Please also make clear that this is a qualitative study.

Ethical statement: “This was part of a mixed-methods study…” If another section of the study has been published (the quantitative portion), then would cite here.

Methods: Were these focus group conducted on online? Please make this clearer (i.e., it’s not clear at first if they were physically conducted in Harare or with people residing in Harare but online).

Methods: Please explain what is means to be “currently engaged in HIV/SRH programs” – does this mean that the participant is a client of these programs? Or has attended these programs at least once? It’s also not clear what the DREAMS program is. Please clarify for an international audience.

Methods: Were participants paid for their time? If yes, please state how much reimbursement/vouchers, etc.

Methods: Please clarify what the criteria for participation in this study is quite broad? Were all participants at a high risk of HIV and hence, should/would know about PrEP?

Methods: Are vaginal rings and CAB-LA available in Zimbabwe? If yes, please add to Introduction.

Methods: Who transcribed the interviews? Were the transcripts de-identified?

Methods: Did the focus groups follow an interview guide? If yes, what sort of questions were on the interview guide?

Methods: The analysis is not clear. The authors describe the coding process and discussions, but how did the authors go about the analysis? How were the results grouped into themes?

Results: It’s not clear from the Table. Did the participants have a stable partner (e.g., boyfriend)?

Results: “transactional sexual relationships” – Could the authors briefly elaborate on this a bit more. Is this a result from the sex workers? Or from all participants? Or are the participants in relationships with the “blessers”?

Results: “This lack of understanding was most pronounced among AGYM…” – Among the 5 participants currently pregnant? Please explain.

Results: Would spell out SRH throughout (and avoid the use of most acronyms to be help increase clarity among an international audience).

Results: “Unfriendly and unprofessional health care worker attitudes…” “Judgmental, unfriendly and unprofessional attitudes by some HCWs…” Is this based on experience or are the participants anticipating that when accessing PrEP, they would be judged. Please clarify.

Results: “pay a consultation fee of $5…” – I think this should be mentioned in the Introduction instead (to help give the reader some context). So is PrEP only free in certain public health facilities and not others?? How many donor funded institutions are in the country? How many people do they service?

Discussion: “Unmarried AGYM…had the least knowledge…” Least knowledge of PrEP? Please provide explain.

Discussion: “This study contributes to the growing literature…” Is this the first study on barriers to PrEP that has taken place in Zimbabwe? If yes, would mention here.

Discussion: “rattling sound from the bottles”? Was this mentioned in the Results? Please clarify.

Discussion: “discreet, fun and engaging” – Could also state that it’s important to change to prevent stigmatization from others (and cite research to indicate this the harms of stigma towards people with HIV and/or people who take PrEP).

Discussion: Would also spell out HCWs throughout the manuscript. Would consider reducing the use of most acronyms in the article

Discussion: Would highlight more in the Methods section how the study involved participatory qualitative methods (In what way is the study participatory?).

Discussion: What were some of the facilitators to PrEP uptake? It seemed that most Results related to barriers?

Discussion: “used participatory approaches…” Again, it’s not clear how the researchers/Methods were participatory? Please explain more in the Methods.

Discussion: A limitation might be the broad participant group? (i.e., as mentioned, included people who had never used PrEP). Please clarify in the Methods if all participants were at high risk of HIV.

Discussion: Please state/clarify in the Methods the % of participants in the study who had experience using PrEP.

Discussion: I think more is needed in this section of what can be done to increase awareness of acceptability of PrEP in Zimbabwe. Are there any global studies that have conducted interventions/programs that have helped to increase the acceptability of PrEP?

Discussion: What future studies are needed in this area? What's next/would be helpful?

**Do you want your identity to be public for this peer review?** For information about this choice, including consent withdrawal, please see our Privacy Policy

Reviewer #1: No

Reviewer #2: No

---

## [Editor Report · Decision Letter 1]

28 Oct 2025

Perceptions of pre-exposure prophylaxis among sexually active adolescent girls and young women in Zimbabwe – a qualitative study.

PGPH-D-25-00857R1

Dear Ms Chidhanguro,

We are pleased to inform you that your manuscript 'Perceptions of pre-exposure prophylaxis among sexually active adolescent girls and young women in Zimbabwe – a qualitative study.' has been provisionally accepted for publication in PLOS Global Public Health.

Best regards,

Guillaume Fontaine, PhD, RN

Academic Editor